# Cardiovascular disease and subsequent risk of psychiatric disorders: a nationwide sibling-controlled study

Qing Shen[1,2]*, Huan Song[1,2,3,4], Thor Aspelund[2], Jingru Yu[5], Donghao Lu[1,3,6], Jóhanna Jakobsdóttir[2], Jacob Bergstedt[1], Lu Yi[5], Patrick Sullivan[5,7], Arvid Sjölander[5], Weimin Ye[5], Katja Fall[1,8], Fang Fang[1†], Unnur Valdimarsdóttir[1,2,6†]

[1]Unit of Integrative Epidemiology, Institute of Environmental Medicine, Karolinska Institutet, Stockholm, Sweden; [2]Centre of Public Health Sciences, Faculty of Medicine, University of Iceland, Reykjavík, Iceland; [3]West China Biomedical Big Data Center, West China Hospital, Sichuan University, Chengdu, China; [4]Medical Big Data Center, Sichuan University, Chengdu, China; [5]Department of Medical Epidemiology and Biostatistics, Karolinska Institutet, Stockholm, Sweden; [6]Department of Epidemiology, Harvard T.H. Chan School of Public Health, Boston, United States; [7]Departments of Genetics and Psychiatry, University of North Carolina, Chapel Hill, United States; [8]Clinical Epidemiology and Biostatistics, School of Medical Sciences, Örebro University, Örebro, Sweden

## Abstract

**Background:** The association between cardiovascular disease (CVD) and selected psychiatric disorders has frequently been suggested while the potential role of familial factors and comorbidities in such association has rarely been investigated.

**Methods:** We identified 869,056 patients newly diagnosed with CVD from 1987 to 2016 in Sweden with no history of psychiatric disorders, and 910,178 full siblings of these patients as well as 10 individually age- and sex-matched unrelated population controls ($N$ = 8,690,560). Adjusting for multiple comorbid conditions, we used flexible parametric models and Cox models to estimate the association of CVD with risk of all subsequent psychiatric disorders, comparing rates of first incident psychiatric disorder among CVD patients with rates among unaffected full siblings and population controls.

**Results:** The median age at diagnosis was 60 years for patients with CVD and 59.2% were male. During up to 30 years of follow-up, the crude incidence rates of psychiatric disorder were 7.1, 4.6, and 4.0 per 1000 person-years for patients with CVD, their siblings and population controls. In the sibling comparison, we observed an increased risk of psychiatric disorder during the first year after CVD diagnosis (hazard ratio [HR], 2.74; 95% confidence interval [CI], 2.62–2.87) and thereafter (1.45; 95% CI, 1.42–1.48). Increased risks were observed for all types of psychiatric disorders and among all diagnoses of CVD. We observed similar associations in the population comparison. CVD patients who developed a comorbid psychiatric disorder during the first year after diagnosis were at elevated risk of subsequent CVD death compared to patients without such comorbidity (HR, 1.55; 95% CI, 1.44–1.67).

**Conclusions:** Patients diagnosed with CVD are at an elevated risk for subsequent psychiatric disorders independent of shared familial factors and comorbid conditions. Comorbid psychiatric disorders in patients with CVD are associated with higher risk of cardiovascular mortality suggesting that surveillance and treatment of psychiatric comorbidities should be considered as an integral part of clinical management of newly diagnosed CVD patients.

*For correspondence:
qing.shen@ki.se

†These authors contributed equally to this work

Competing interest: The authors declare that no competing interests exist.

**Funding:** This work was supported by the EU Horizon 2020 Research and Innovation Action Grant (CoMorMent, grant no. 847776 to UV, PFS, and FF), Grant of Excellence, Icelandic Research Fund (grant no. 163362-051 to UV), ERC Consolidator Grant (StressGene, grant no. 726413 to UV), Swedish Research Council (grant no. D0886501 to PFS), and US NIMH R01 MH123724 (to PFS).

## Editor's evaluation

Whether a diagnosis of cardiovascular disease (CVD) increases risks of psychiatric disorders is not well understood. Using a large population-based case-sibling study in Sweden, this important study suggests that patients diagnosed with CVD are at higher risk of psychiatric disorders, independent of familial factors shared between full siblings and comorbid conditions. While the analysis is solid, additional variables that relate to both CVD risk and mental health need to be incorporated in follow-up analyses.

## Introduction

Being diagnosed and living with a major life-threatening disease is stressful and associated with multiple biologic processes that, when combined, may contribute to the development of psychiatric disorders. It is for instance demonstrated that a cancer diagnosis is associated with subsequent risk of psychiatric disorders (*Lu et al., 2016*) and self-inflicted injury (*Shen et al., 2016*), which in turn might be associated with a compromised cancer survival (*Zhu et al., 2017*). Psychiatric comorbidities have also been reported among patients with cardiovascular disease (CVD), for example, stroke (*Lindén et al., 2007*), heart failure (*Rutledge et al., 2006*), and myocardial infarction (*Thombs et al., 2006*; *Shemesh et al., 2004*), with indications of elevated risk of overall mortality (*Doering et al., 2010*; *Wrenn et al., 2013*). Yet, evidence on the association between CVD and subsequent development of psychiatric disorders is still limited as previous research has mainly relied on selected patient populations instead of complete follow-up of general population as well as limited control of reverse causality and important confounding factors, for example, familial factors and comorbidities (*Lincoln et al., 2013*; *Morrison et al., 2005*; *Romanelli et al., 2002*).

We thoroughly searched the existing literature on the association between CVD and clinically confirmed psychiatric disorders or psychiatric symptoms. After excluding most studies with either cross-sectional or retrospective designs, we only found 12 prospective cohort studies investigating the risk of selected psychiatric disorders following a diagnosis of CVD (*Supplementary file 1a*). While these prospective studies suggest a positive association of hypertension, heart disease, and stroke with depressive symptoms (*Morrison et al., 2005*; *Wium-Andersen et al., 2017*; *Baccaro et al., 2019*; *Pohjasvaara et al., 2001*) and stress-related disorders (*Chang et al., 2017*), they were limited to elderly populations (*Petersson et al., 2014*; *Zhang et al., 2015*), and used self-reported ascertainment of psychiatric outcomes (*Petersson et al., 2014*; *Zhang et al., 2015*) Only few of these studies addressed incident or first diagnosed psychiatric disorders among patients with CVD, for example, excluding patients with history of psychiatric disorders, and no study addressed the issue of familial confounding. Indeed, genetic correlation has recently been document between these two complex disease groups (*Barnett and Smoller, 2009*; *Kathiresan and Srivastava, 2012*; *Rødevand et al., 2021*) as well as the importance of early life environment for the development of both CVD (*Kelishadi and Poursafa, 2014*) and psychiatric disorders (*Rokita et al., 2018*). It is therefore unknown to what extent the reported association between CVD and psychiatric disorder can be explained by unmeasured confounding shared within families (*Barnett and Smoller, 2009*; *Kathiresan and Srivastava, 2012*; *Rødevand et al., 2021*). Thus, a comprehensive evaluation of the association between all CVDs and risk of any incident psychiatric disorder, addressing the abovementioned shortcomings, is warranted.

With up to 30 years of follow-up and with nationwide complete information on family links in Sweden, we aimed to investigate the association between CVD diagnosed in specialist care and subsequent risk of incident psychiatric disorders while accounting for familial factors through a sibling comparison. We further aimed to estimate the potential role of psychiatric comorbidity in cardiovascular mortality among patients with CVD.

## Materials and methods

### Study design

The Swedish Patient Register contains national information on inpatient care with complete coverage since 1987 and outpatient specialized care since 2001 (*Ludvigsson et al., 2011*). The Swedish Multi-Generation Register includes nearly complete familial information for Swedish residents born since 1932 (*Ekbom, 2011*). Using personal identification numbers assigned to all Swedish residents, we identified all individuals born in Sweden after 1932 who received a first diagnosis of any CVD and attended inpatient or outpatient specialized care between January 1, 1987 and December 31, 2016 (*N* = 986,726). Patients diagnosed with CVD before age 5 (*N* = 6091, probable congenital heart disease) or with a history of any psychiatric disorder before the diagnosis of CVD (*N* = 111,579) were excluded, leaving 869,056 patients in the analysis (*Supplementary file 2*). Date of first CVD diagnosis was used as the index date for the exposed patients.

We constructed a sibling-controlled matched cohort to control for familial confounding according to guidelines for designing family-based studies (*D'Onofrio et al., 2013*). Through the Multi-Generation Register, we identified all full siblings of patients with CVD (58.6% of all CVD patients) who were alive and free of CVD and psychiatric disorder at the time when their affected sibling was diagnosed (*N* = 910,178). In addition, for each patient with CVD, we randomly selected 10 age- and sex-matched individuals from the general population who were free of CVD or psychiatric disorder when the index patient was diagnosed (*N* = 8,690,560). The date of CVD diagnosis for the index patient was used as the index date for their unaffected siblings and matched population controls.

All study participants were followed from the index date until first diagnosis of any psychiatric disorder, death, emigration, first diagnosis of CVD (for unaffected siblings and matched population controls), or the end of the study period (December 31, 2016), whichever occurred first.

### Ascertainment of CVD and psychiatric disorder

We defined CVD as any first inpatient or outpatient hospital visit with CVD as the primary diagnosis from the Swedish Patient Register. Incident psychiatric disorder was defined as any first inpatient or outpatient hospital visit with psychiatric disorder as the primary diagnosis. We used the 9th and 10th Swedish revisions of the International Classification of Diseases (ICD-9 and 10) codes to identify CVD and psychiatric disorders and their subtypes (*Supplementary file 1b*). In line with previous study (*Song et al., 2019*), we classified CVD as ischemic heart disease, cerebrovascular disease, emboli/thrombosis, hypertensive disease, heart failure, and arrhythmia/conduction disorder. We classified psychiatric disorders as non-affective psychotic disorders, affective psychotic disorders, alcohol or drug misuse, mood disorders excluding psychotic symptoms, anxiety and stress-related disorders, eating disorders, and personality disorders (*Nevriana et al., 2020*).

### Covariates

We extracted socioeconomic information for each participant, including educational level, individualized family income, and cohabitation status, from the Longitudinal Integration Database for Health Insurance and Labor Market (*Longitudinell integrationsdatabas för sjukförsäkrings- och arbetsmarknadsstudier (LISA), 2018*). Missing information on socioeconomic status was categorized as unknown or missing group. A history of somatic diseases was defined as having any of the following conditions before the index date: chronic pulmonary disease, connective tissue disease, diabetes, renal diseases, liver disease, ulcer diseases, malignancies, and HIV infection/AIDS (*Supplementary file 1b*). We defined a family history of psychiatric disorders as a diagnosis of any psychiatric disorder among biological parents and full siblings of the study participants before the index date according to the Swedish Patient Register.

### Statistical analysis

We used flexible parametric survival models to estimate the time-varying association between CVD and subsequent risk of incident psychiatric disorders (*Lambert and Royston, 2009*), by comparing the rates of incident psychiatric disorders in CVD patients with the corresponding rates in their unaffected full siblings and matched population controls. As we observed a marked risk increase of psychiatric disorders immediately following the CVD diagnosis, we separately assessed the association within 1 year of CVD diagnosis and beyond 1 year. Hazard ratios (HRs) and their 95% confidence intervals (CIs)

were derived from stratified Cox regression models, using time since the index date as the underlying time scale. We estimated HRs for any psychiatric disorder and categories of psychiatric disorders. We performed subgroup analyses by sex, age at index date (<50, 50–60, or >60 years), age at follow-up (<60 or ≥60 years), history of somatic diseases (no or yes), and family history of psychiatric disorder (no or yes). We also performed subgroup analysis by calendar year at index date (1987–1996, 1997–2006, or 2007–2016) to check for potentially different associations over time (i.e., due to lifestyle factors that changed over time, including smoking and alcohol use) (*Sundin and Willner, 2007*). In the sibling comparison, all Cox models were stratified by sibling sets and adjusted for sex, birth year, educational level, individualized family income, cohabitation status, and history of somatic diseases. In the population comparison, all Cox models were stratified by the matching variables birth year and sex and adjusted for all abovementioned covariates plus family history of psychiatric disorder.

To study the impact of additional cardiovascular comorbidity (i.e., patients with another type of CVD after diagnosis of the first CVD), we analyzed the association by the presence or absence of cardiovascular comorbidity after the index date according to the type of first CVD. This analysis was restricted to follow-up beyond 1 year to focus on patients who survived their first CVD to be able to receive the diagnosis of another CVD. As a patient with CVD might have different types of CVD, we identified all diagnoses of CVD during follow-up and considered CVD comorbidity as a time-varying variable through splitting the person-time according to each diagnosis.

Because the Swedish Patient Register includes only information related to specialist care, we might have misclassified patients with a history of milder psychiatric disorders diagnosed before index date attended only in primary care. To account for the reverse causality of having undetected psychiatric disorders or symptoms before the incident CVD, we performed a sensitivity analysis additionally excluding study participants with prescribed use of psychotropic drugs before the index date (ascertained from the Swedish Prescribed Drug Register including information on all prescribed medication use in Sweden since July 2005), and followed the remaining participants during 2006–2016. Use of psychotropic drugs during follow-up was also considered as having psychiatric disorder in this analysis.

To study rate of cardiovascular mortality (ascertained from the Swedish Causes of Death Register) in relation to psychiatric comorbidities after CVD diagnosis, we estimated Kaplan–Meier survival curves beyond the first year of follow-up for CVD patients with or without a diagnosis of psychiatric disorder during the first year of follow-up, separately. We estimated the survival curves by types of first diagnosed CVD as well as by types of psychiatric comorbidities. In addition to this 1-year time window, we also studied 6 months or 2 years since CVD diagnosis, to assess the robustness of these survival curves. We calculated the HRs of cardiovascular mortality for these two groups of patients using Cox model. To account for potential impact of unmeasured confounding due to lifestyle factors, we performed a sensitivity analysis excluding individuals with a history of alcoholic cirrhosis of liver (ICD-10 code K703) or chronic obstructive pulmonary disease (COPD, ICD-10 code J44), as proxies for heavy drinking or smoking.

Analyses were performed in STATA 17.0 (StataCorp LP). All tests were two sided and p < 0.05 was considered statistically significant. The study was approved by the Ethical Vetting Board in Stockholm, Sweden (DNRs 2012/1814-31/4 and 2015/1062-32).

### Role of the funding source

The funders of the study had no role in study design, data collection, data analysis, data interpretation, or writing of the report.

## Results

The median age at index date was 60 years for CVD patients and 55 years for their unaffected full siblings (*Table 1*). 59.2% of the CVD patients and 48.4% of their unaffected siblings were male. CVD patients were more likely to have a history of somatic diseases than their unaffected siblings and matched population controls (15.6% vs. 8.8% and 11.0%). The most common diagnoses among the CVD patients were ischemic heart diseases (24.5%), arrhythmia/conduction disorders (24.2%), and hypertensive diseases (17.3%). The majority of the CVD patients had only one CVD diagnosis (without additional CVD comorbidities) during follow-up (69.7%).

**Table 1.** Characteristics of CVD patients diagnosed in Sweden between 1987 and 2016, their unaffected siblings and matched population controls.

| Characteristics | Sibling comparison | | Population comparison | |
| --- | --- | --- | --- | --- |
| | CVD patients (*N* = 509,467) | Unaffected full siblings (*N* = 910,178) | CVD patients (*N* = 869,056) | Matched population controls (*N* = 8,690,560) |
| Median age at index date in years (IQR) | 57 (48–65) | 55 (46–63) | 60 (51–68) | 60 (51–68) |
| Median follow-up time in years (IQR) | 8.1 (3.7–13.7) | 8.1 (3.8–13.7) | 7.7 (3.3–13.2) | 7.1 (3.2–12.4) |
| Male sex | 308,203 (60.5) | 440,177 (48.4) | 514,388 (59.2) | 5,143,880 (59.2) |
| Educational level | | | | |
| <9 years | 149,555 (29.4) | 261,752 (28.8) | 272,960 (31.4) | 2,294,482 (26.4) |
| 9–12 years | 225,548 (44.3) | 413,702 (45.5) | 376,917 (43.4) | 3,548,338 (40.8) |
| >12 years | 134,364 (26.4) | 234,724 (25.8) | 219,179 (25.2) | 2,847,740 (32.8) |
| Yearly individualized family income level | | | | |
| Top 20% | 107,990 (21.2) | 175,658 (19.3) | 139,098 (16.0) | 1,757,726 (20.2) |
| Middle | 301,706 (59.2) | 549,842 (60.4) | 535,109 (61.6) | 5,152,938 (59.3) |
| Lowest 20% | 99,485 (19.5) | 184,588 (20.3) | 192,858 (22.2) | 1,706,931 (19.6) |
| Unknown | 286 (0.1) | 90 (0.0) | 1991 (0.2) | 72,965 (0.8) |
| Cohabitation status | | | | |
| Non-cohabitating | 223,134 (43.8) | 392,256 (43.1) | 373,337 (43.0) | 3,744,116 (43.1) |
| Cohabitating | 286,047 (56.2) | 517,832 (56.9) | 493,728 (56.8) | 4,873,479 (56.1) |
| Missing | 286 (0.1) | 90 (0.0) | 1991 (0.2) | 72,965 (0.8) |
| History of somatic disease* | 71,273 (14.0) | 79,679 (8.8) | 135,473 (15.6) | 955,030 (11.0) |
| Family history of psychiatric disorder† | 133,094 (26.1) | 251,237 (27.6) | 209,957 (24.2) | 2,003,161 (23.1) |
| Type of first-onset CVD | | | | |
| Ischemic heart disease | 122,084 (24.0) | – | 212,737 (24.5) | – |
| Cerebrovascular disease | 71,030 (13.9) | – | 126,860 (14.6) | – |
| Emboli and thrombosis | 25,338 (5.0) | – | 42,857 (4.9) | – |
| Hypertensive disease | 89,818 (17.6) | – | 150,337 (17.3) | – |
| Heart failure | 15,726 (3.1) | – | 30,469 (3.5) | – |
| Arrhythmia/conduction disorder | 126,738 (24.9) | – | 210,654 (24.2) | – |
| Others | 58,733 (11.5) | – | 95,142 (11.0) | – |
| Number of cardiovascular diagnoses during follow-up | | | | |
| One | 365,266 (71.7) | – | 605,615 (69.7) | – |
| Two | 99,921 (19.6) | – | 179,472 (20.7) | – |
| Three or more | 44,280 (8.7) | – | 83,969 (9.7) | – |

*History of somatic diseases included chronic pulmonary disease, connective tissue disease, diabetes, renal diseases, liver diseases, ulcer diseases, and HIV infection/AIDS that diagnosed before index date.

†The difference between exposed patients and unaffected full siblings was due to different number of siblings for exposed patients. The family history of psychiatric disorder was constant within each family.

IQR: interquartile range. CVD: cardiovascular disease.

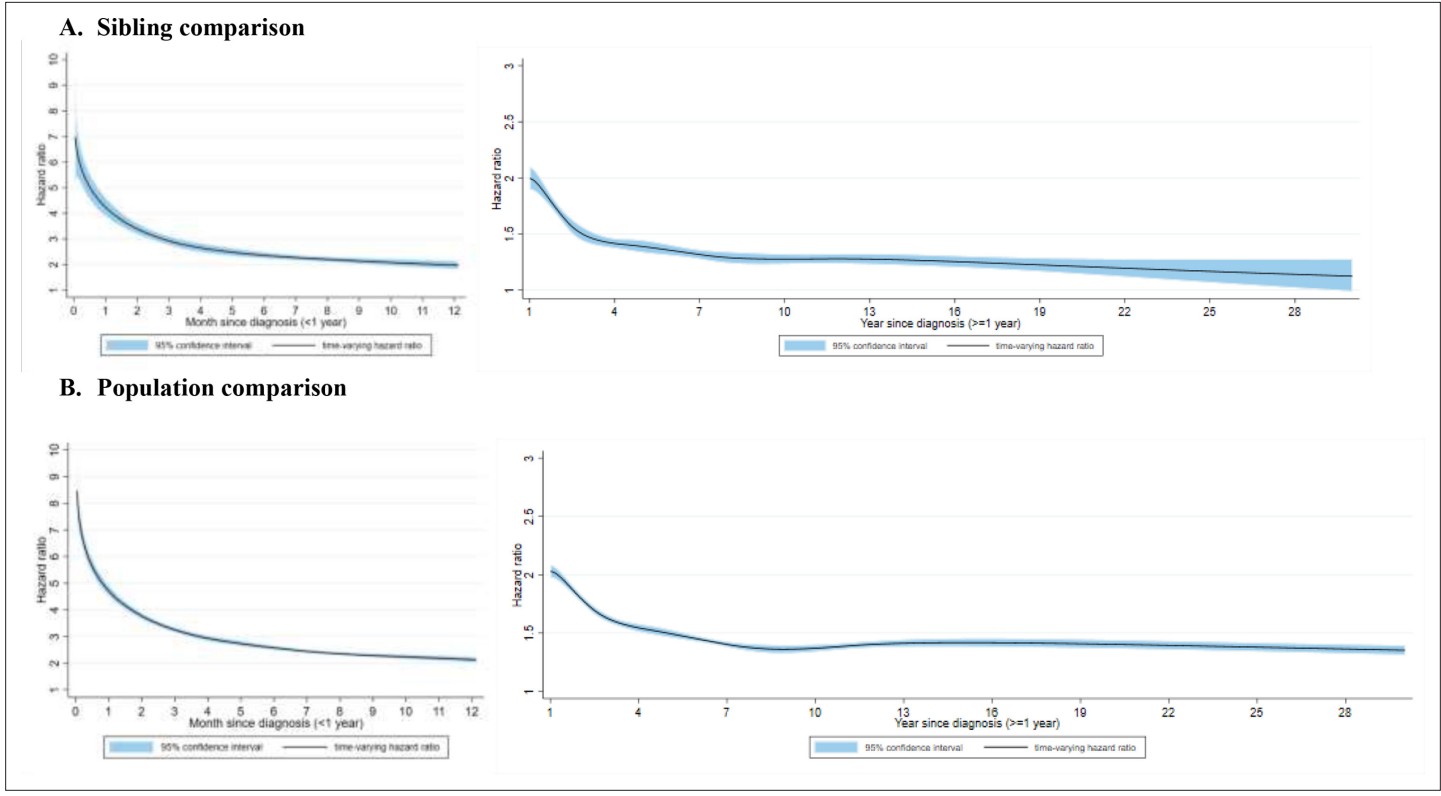

**Figure 1.** Time-varying hazard ratios for an incident psychiatric disorder among CVD patients, compared with their unaffected full siblings (sibling comparison) or matched population controls (population comparison), by time of follow-up (<1 and ≥1 year from CVD diagnosis)*. (A) Sibling comparison. (B) Population comparison. *CVD: cardiovascular disease. Time-varying hazard ratios and 95% confidence intervals were derived from flexible parametric survival models, allowing the effect of psychiatric disorder to vary over time. A spline with 5 df was used for the baseline rate, and 3 df was used for the time-varying effect. All models were adjusted for age at index date, sex, educational level, yearly individualized family income, cohabitation status, history of somatic diseases, as well as family history of psychiatric disorder (for population comparison). P<0.05 was considered level of significance.

During up to 30 years of follow-up, the crude incidence rates of psychiatric disorder were 7.1, 4.6, and 4.0 per 1000 person-years among CVD patients, their unaffected full siblings, and matched population controls, respectively (*Supplementary file 1c*). Compared with unaffected siblings, CVD patients showed an elevated risk of incident psychiatric disorder, especially immediately after diagnosis (*Figure 1*). The risk increase declined rapidly within the first few months after diagnosis and decreased gradually thereafter: the HR was 2.74 (95% CI, 2.62–2.87) within first year and 1.45 (95% CI, 1.42–1.48) beyond first year (*Supplementary file 1d*). The risk increment was noted in all types of psychiatric disorders within and beyond first year of follow-up (*Figure 2* and *Supplementary file 1e*). Overall, the observed positive association was similar in sibling and population comparisons, although the HR of non-affective psychotic disorders beyond 1 year of CVD diagnosis was smaller in the sibling comparison than in the population comparison. During the entire follow-up, we found similar positive associations across sex, age at index date, age at follow-up, history of somatic diseases, and family history of psychiatric disorders (*Supplementary file 1c*). A greater risk increment was observed in recent calendar years than earlier years.

We found a higher risk of incident psychiatric disorder among all groups of CVD patients, with the most marked risk elevation observed among patients with cerebrovascular disease and heart failure (*Figure 3* and *Supplementary file 1f*). A greater risk increment of incident psychiatric disorder was noted among CVD patients with additional cardiovascular comorbidities beyond 1 year of first CVD diagnosis compared with CVD patients without such comorbidities, except among those with heart failure (*Figure 3—figure supplement 1*). We found a similar positive association between CVD and

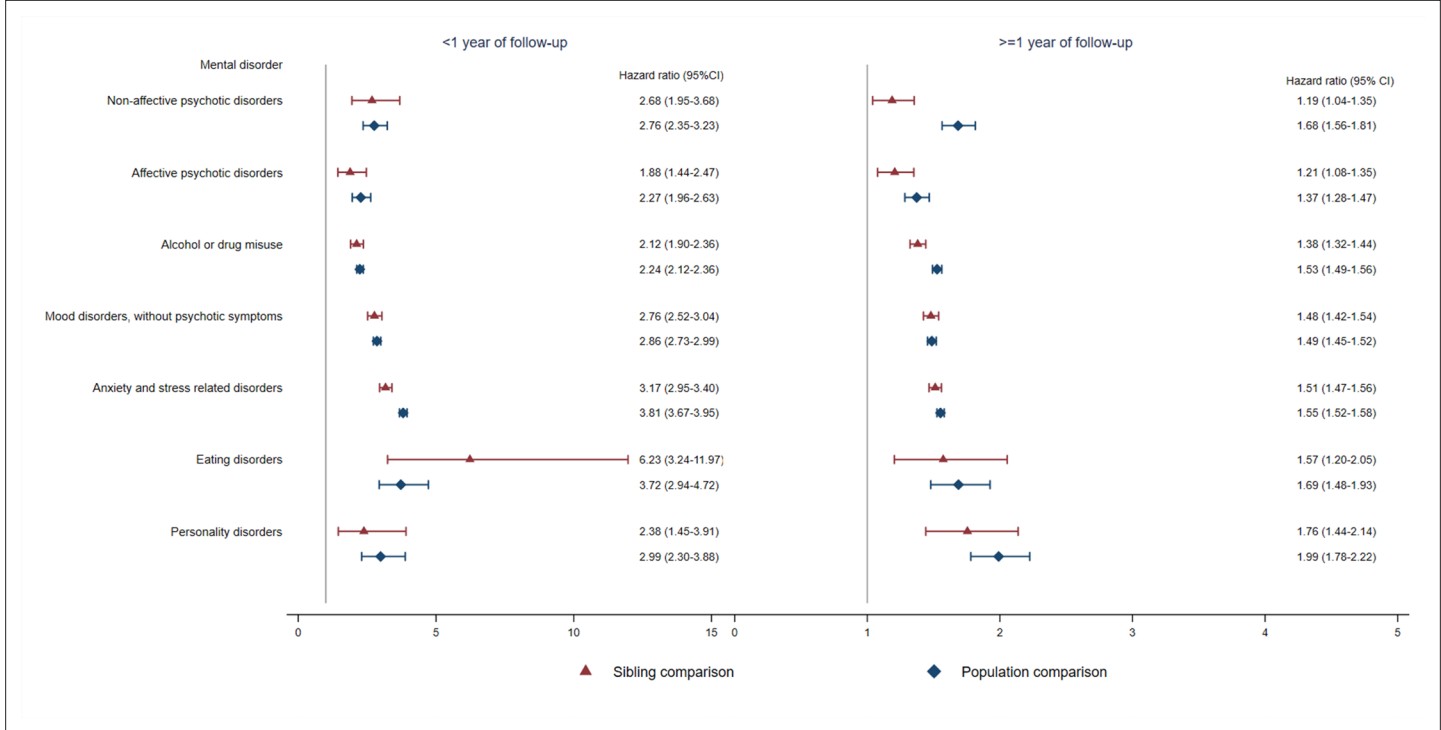

**Figure 2.** Hazard ratios with 95% confidence intervals for different types of psychiatric disorder among CVD patients compared with their full siblings and matched population controls, by time of follow-up (<1 or ≥ 1 year from CVD diagnosis)*. *CVD: cardiovascular disease. Cox regression models were stratified by family identifier for sibling comparison or matching identifier (birth year and sex) for population comparison, controlling for age at index date, sex, educational level, individualized family income, cohabitation status, history of somatic diseases, and family history of psychiatric disorder (in population comparison). Time since index date was used as underlying time scale. P<0.05 was considered level of significance.

risk of incident psychiatric disorder, when including use of psychotropic drugs as a definition of psychiatric disorder (**Supplementary file 1g**).

CVD patients diagnosed with subsequent psychiatric disorder showed a lower CVD-specific survival compared with patients without such diagnosis (**Figure 4** and **Figure 4—figure supplement 1**). The HR of CVD death was 1.55 (95% CI 1.44–1.67) when comparing CVD patients with a diagnosis of psychiatric disorder to patients without such diagnosis (mortality rate, 9.2 and 7.1 per 1 000 person-years, respectively). The compromised CVD-specific survival differed by types of CVD, and was most pronounced for hypertensive disease, ischemic heart disease, and arrhythmia/conduction disorder (**Figure 4—figure supplement 2**). When studying categories of psychiatric disorders, we found that the compromised survival among CVD patients was confined to those with comorbid non-affective and affective psychotic disorders, as well as alcohol or drug misuse (**Figure 4—figure supplement 3**).

## Discussion

Our large population-based sibling-controlled cohort study including all patients diagnosed with first-onset CVD between 1987 and 2016 in Sweden, their unaffected full siblings, as well as a set of randomly selected unaffected population controls reveals a robust association between CVD and subsequent risk of incident psychiatric disorder. We found that patients with CVD were at elevated risk of various types of psychiatric disorders, independent of confounding factors shared within families and history of somatic diseases. The risk increment was greatest during the year after CVD diagnosis, indicating an opportunity for clinical surveillance in a high-risk time window. Further, an occurrence of psychiatric comorbidity after CVD diagnosis was associated with an approximately 55% higher risk of subsequent death from cardiovascular causes. This finding further underscores the importance of surveillance and, if needed, treatment of psychiatric comorbidities among newly diagnosed CVD patients.

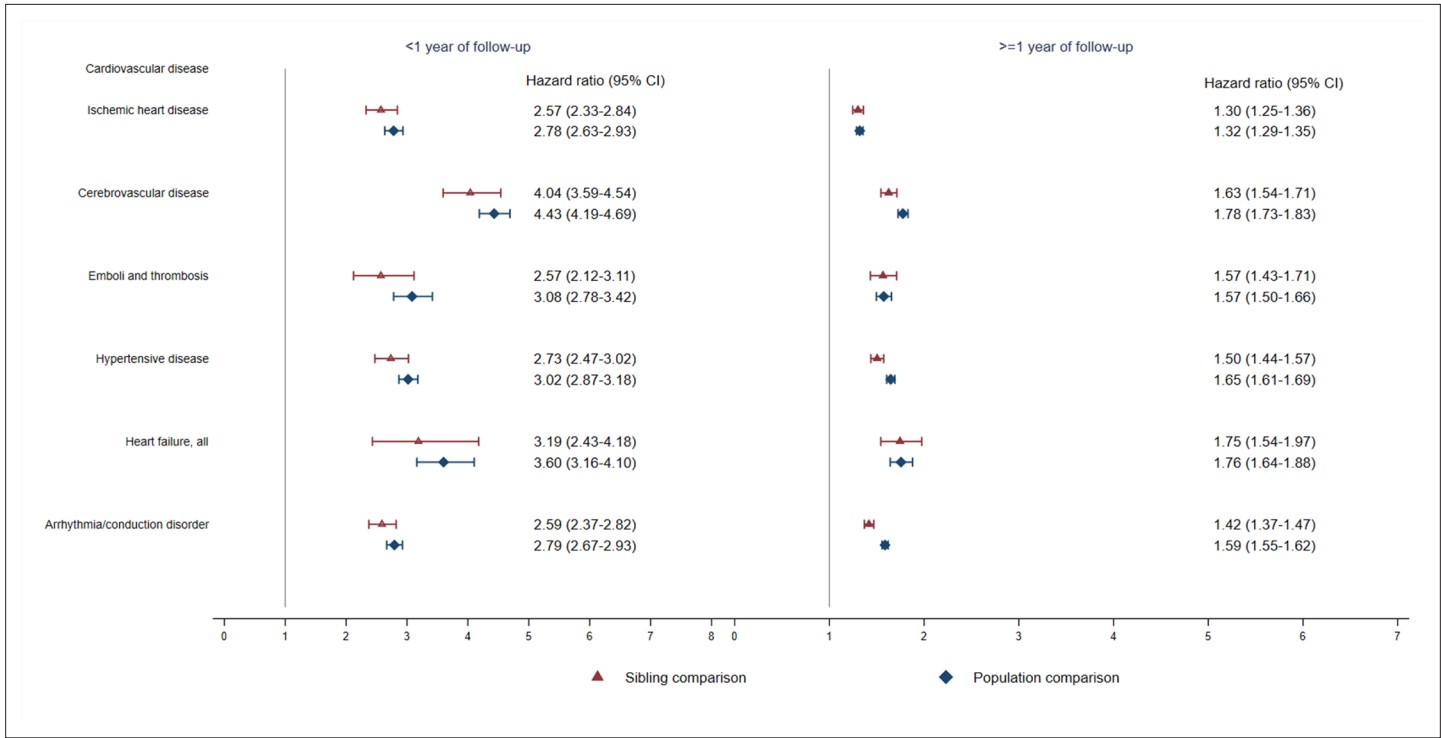

**Figure 3.** Hazard ratios with 95% confidence intervals for psychiatric disorders among different groups of CVD patients compared with their full siblings and matched population controls, by time of follow-up (<1 or ≥ 1 year from CVD diagnosis)*. *CVD: cardiovascular disease. Cox regression models were stratified by family identifier for sibling comparison or matching identifier (birth year and sex) for population comparison, controlling for age at index date, sex, education level, individualized family income, cohabitation status, history of somatic diseases, and family history of psychiatric disorder (in population comparison). Time since index date was used as underlying time scale. We identified all cardiovascular diagnoses during follow-up and considered CVD comorbidity as a time-varying variable by grouping the person-time according to each diagnosis. P<0.05 was considered level of significance.

The online version of this article includes the following figure supplement(s) for figure 3:

**Figure supplement 1.** Crude incidence rates and hazard ratios with 95% confidence intervals (CIs) for an incident psychiatric disorder among different types of CVD patients compared with their full siblings or matched population controls, by number of CVD diagnoses during ≥1 year of follow-up[a].

## Comparison with other studies

Our findings are consistent with the existing literature suggesting a positive association between CVD and different types of psychiatric disorders. In previous studies, an increased risk of depression (*Wium-Andersen et al., 2017*; *Baccaro et al., 2019*; *Pohjasvaara et al., 2001*) and anxiety (*Morrison et al., 2005*) was noted after diagnosis of stroke (*Morrison et al., 2005*; *Wium-Andersen et al., 2017*; *Baccaro et al., 2019*; *Pohjasvaara et al., 2001*), hypertension (*Petersson et al., 2014*), coronary artery disease (*Rutledge et al., 2013*), and atrial fibrillation (*Baumgartner et al., 2018*). Such risk elevations have been suggested to be persistent over time (*Lincoln et al., 2013*; *Zawadzka and Domańska, 2014*; *Berg et al., 2003*), and, in parallel with our findings, associated with compromised survival (*Doering et al., 2010*; *Cai et al., 2019*; *Bodén et al., 2015*). However, the evidence from prospective cohort studies with a long and complete follow-up as well as with a thorough control of confounding factors and comorbid conditions has, up to this point, been limited. Our study therefore complements previous findings revealing a positive association between a broader range of CVDs and subsequent risk of incident psychiatric disorder, using a large cohort with control of various confounding factors. We found the association to be robust both in sibling and population comparisons, and after additional adjustment for various comorbidities including other additional CVDs, indicating that the association is unlikely explained by shared familial factors and various comorbidities. In addition to common psychiatric disorders, the evidence on CVD and other psychiatric disorders, for example, eating disorder, is limited. Our study therefore provides valuable indication on this association that deserves further research attention. We showed that CVD patients with psychiatric comorbidity were

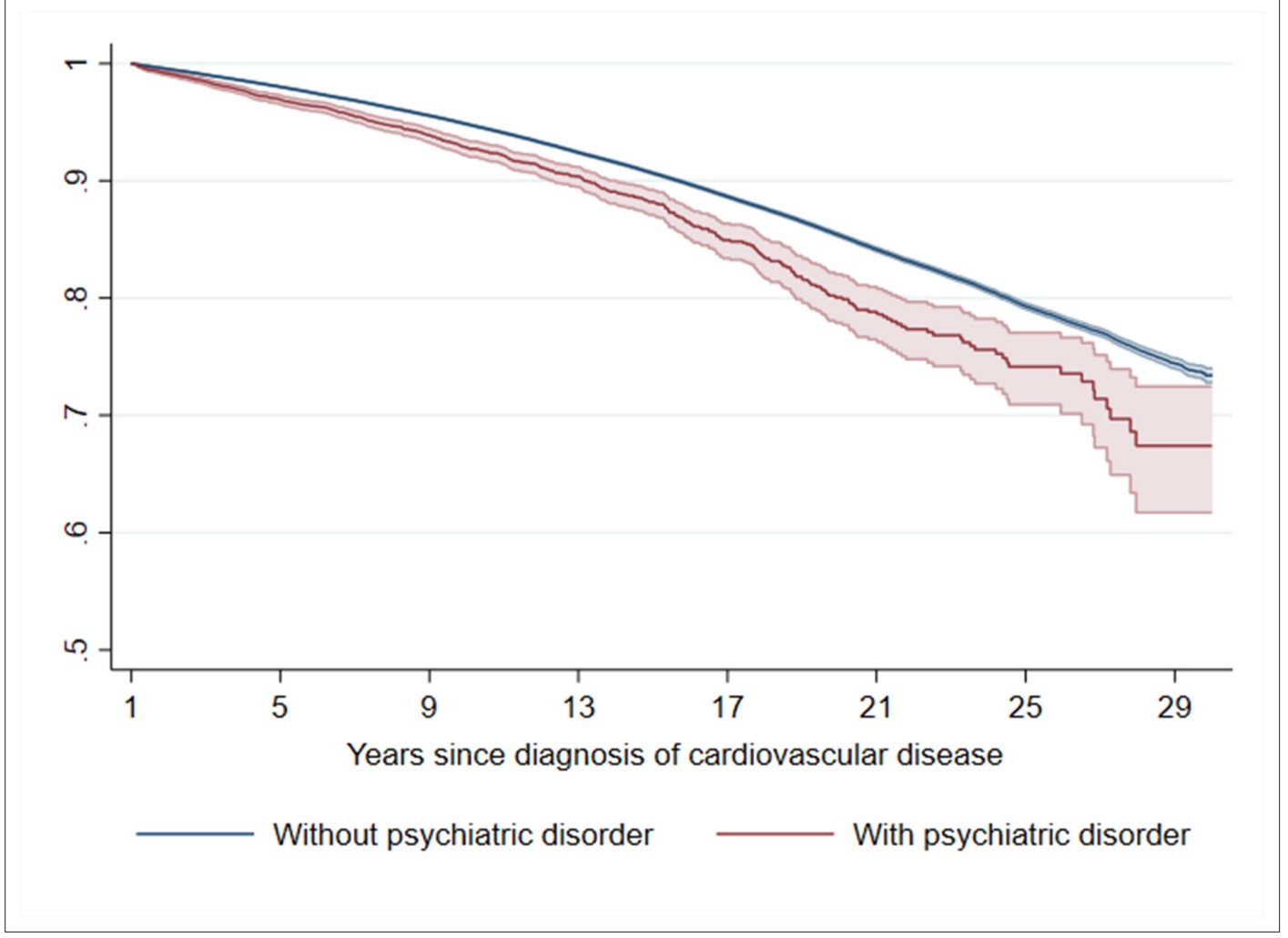

**Figure 4.** Estimated Kaplan–Meier curves of CVD death in CVD patients with and without incident psychiatric disorder during the first year of follow-up[a].
[a]CVD: cardiovascular disease. Time since index date was used as underlying time scale. 90.4% of CVD patients (*N* = 785,287) survived the first year of follow-up and included in this analysis. P<0.05 was considered level of significance.

The online version of this article includes the following figure supplement(s) for figure 4:

**Figure supplement 1.** Estimated Kaplan–Meier curves of CVD death in CVD patients with and without incident psychiatric disorder during (1) 6 months or (2) 2 years of follow-up.

**Figure supplement 2.** Estimated Kaplan–Meier curves of CVD death in CVD patients with and without incident psychiatric disorder during the first year of follow-up, according to types of first CVD diagnosis[a].

**Figure supplement 3.** Estimated Kaplan–Meier curves of CVD death in CVD patients with and without incident psychiatric disorder during the first year of follow-up, according to types of incident psychiatric disorder[a].

associated with an increased risk of subsequent CVD death, highlighting the importance of surveillance and prevention on psychiatric comorbidities for the newly diagnosed CVD patients. The associations remained similar after excluding individuals with liver cirrhosis and COPD, as proxies for heavy alcohol consumption and smoking, suggesting that residual confounding due to unmeasured lifestyle factors might not have overly substantial impact on the results.

The association between CVD and psychiatric disorders was noted both in men and women, across all age groups and calendar periods, as well as among individuals with or without a history of somatic diseases and family history of psychiatric disorders. Previous studies have indicated that major depression was more commonly recognized among individuals with multimorbidity (more than one CVD diagnosis) than those with only one condition (*Findley et al., 2011*). In our study, about 30% of the

CVD patients developed one or more cardiovascular comorbidities during follow-up, and a higher risk of psychiatric disorder was indeed noted among patients with multiple CVD diagnoses. A compromised survival from CVD cause was indeed observed among most patients with common types of first CVD diagnosis and comorbid with psychiatric disorder, in particular among patients with hypertensive disease, ischemic heart disease, and arrhythmia/conduction disorder. Thus, particular clinical attention is needed for CVD patients with comorbid psychiatric disorders, particularly alcohol or drug misuse or psychotic disorders.

## Potential mechanisms

The pathophysiological mechanisms linking CVD and psychiatric disorders are complex and not well understood, and may vary with specific diagnoses of CVD and psychiatric disorders. The highly increased risk noted immediately after CVD diagnosis may indicate a direct impact of stress reaction of being diagnosed with a life-threatening disease (*Fang et al., 2012*). In addition, it has been proposed that biological alterations in the cardiovascular system to a severe stress response may increase the risk of various psychiatric disorders (*Levine et al., 2021*). For example, cardiovascular risk factors including hypercoagulability, dyslipidemia, and an impaired immune response have been associated with impaired psychological health (*Levine et al., 2021*). Some biological changes in patients with coronary heart disease (e.g., decreased heart rate variability, increased arterial stiffness, and endothelial dysfunction) have been observed in patients with depressive and anxiety disorders (*Sherwood et al., 2005*; *Seldenrijk et al., 2011*; *Stein et al., 2000*). Chronic inflammation may induce the development of atherosclerosis and arterial thrombosis, and elevation in inflammatory biomarkers (i.e., IL-6 and C-reactive protein) has been reported in various psychiatric disorders including post-traumatic stress disorder and major depression (*Libby, 2006*; *Sumner et al., 2020*; *Miller and Raison, 2016*). Other behavioral and psychosocial factors may as well interact with these pathways and need to be understood further.

## Strengths and limitations

The strengths of our study include its large sample size of the entire Swedish nation and the prospective study design with sibling comparison that significantly alleviates concerns of familial confounding from shared genetic and environmental factors between siblings. The Swedish population and health registers provide the opportunity to obtain complete follow-up as well as the prospectively and independently collected information on disease identification, minimizing the risk of selection and information biases. The large sample size of our study further enables detailed subgroup analyses by types of CVD, types of psychiatric disorders, and patient characteristics.

Some limitations need to be acknowledged. First, we identified patients with CVD and psychiatric disorder through inpatient or outpatient hospital visit. The later inclusion of outpatient records in the Swedish Patient Register may lead to underestimation of the actual numbers of patients with CVD and psychiatric disorder, especially those with relatively milder symptoms. Second, we missed individuals attending primary care only, which may underestimate the proportion of individuals with history of psychiatric disorders at cohort entry. To alleviate such concerns, we additionally considered the use of prescribed psychotropic drugs as a proxy of psychiatric disorders and found similar results. Further, patients with CVD have an established contact with health care and may therefore be more likely than others to be diagnosed with psychiatric disorder. Although such surveillance bias may to some extent explain the increased risks during the first few months after CVD diagnosis, it is unlikely that the risk elevation during the entire follow-up is attributed to such bias. Finally, although we found similar results with and without excluding individuals with a history of liver cirrhosis or COPD, as proxies for heavy drinking or smoking (*Supplementary file 1h*). We did not have direct access to hazardous behaviors that could potentially modify this association, and therefore cannot exclude the possibility of residual confounding.

## Conclusions

Using a large population-based sibling-controlled cohort with up to 30 years of follow-up, we found patients with CVD are at elevated risk of newly diagnosed psychiatric disorder, independent of familial background shared between siblings, history of somatic diseases, and other cardiovascular comorbidities. Our study further observes higher cardiovascular mortality among CVD patients with subsequent

psychiatric comorbidities, providing evidence for increased surveillance of psychiatric comorbidity among newly diagnosed patients with CVD.

## Additional information

### Funding

| Funder | Grant reference number | Author |
|---|---|---|
| EU Horizon 2020 Research and Innovation Action Grant | CoMorMent,847776 | Patrick Sullivan |
| Grant of Excellence, Icelandic Research Fund | 163362-51 | Unnur Valdimarsdóttir |
| European Research Council | Consolidator Grant StressGene,726413 | Unnur Valdimarsdóttir |
| Swedish Research Council | D0886501 | Patrick Sullivan |
| National Institute of Mental Health | R01 MH123724 | Patrick Sullivan |

The funders had no role in study design, data collection, and interpretation, or the decision to submit the work for publication.

### Author contributions

Qing Shen, Conceptualization, Formal analysis, Investigation, Visualization, Methodology, Writing - original draft, Writing – review and editing; Huan Song, Conceptualization, Investigation, Methodology, Writing – review and editing; Thor Aspelund, Arvid Sjölander, Katja Fall, Investigation, Methodology, Writing – review and editing; Jingru Yu, Data curation, Investigation, Writing – review and editing; Donghao Lu, Jóhanna Jakobsdóttir, Jacob Bergstedt, Lu Yi, Investigation, Writing – review and editing; Patrick Sullivan, Funding acquisition, Investigation, Writing – review and editing; Weimin Ye, Resources, Data curation, Writing – review and editing; Fang Fang, Unnur Valdimarsdóttir, Conceptualization, Supervision, Funding acquisition, Investigation, Methodology, Writing – review and editing

### Author ORCIDs

Qing Shen http://orcid.org/0000-0002-7214-4797
Thor Aspelund http://orcid.org/0000-0002-7998-5433
Donghao Lu http://orcid.org/0000-0002-4186-8661
Fang Fang http://orcid.org/0000-0002-3310-6456
Unnur Valdimarsdóttir http://orcid.org/0000-0001-5382-946X

### Ethics

The study was approved by the Ethical Vetting Board in Stockholm, Sweden (DNRs 2012/1814-31/4 and 2015/1062-32). Informed consent to each participant was waived by Swedish law in nationwide registry data.

### Decision letter and Author response

Decision letter https://doi.org/10.7554/eLife.80143.sa1
Author response https://doi.org/10.7554/eLife.80143.sa2

## Additional files

### Supplementary files

• Supplementary file 1. (a) Summary of prospective cohort studies addressing the association between various indications of cardiovascular disease and risk of psychiatric disorders/psychiatric symptoms. (b) International Classification of Diseases (ICD) codes for exposure, outcome, and covariates identifications. (c) Crude incidence rates (IRs) and hazard ratios (HRs) with 95% confidence intervals (CIs) for incident psychiatric disorder among CVD patients compared with their full siblings or matched population controls, by patient characteristics. CVD: cardiovascular

disease. [a]Cox regression models, stratified by family identifier for sibling comparison or matching identifier (birth year and sex) for population comparison, adjusting for sex, birth year, educational level, individualized family income, cohabitation status, history of somatic disease, and family history of psychiatric disorder. Time since index date was used as underlying time scale. (d) Crude incidence rates (IRs) and hazard ratios (HRs) with 95% confidence intervals (CIs) for incident psychiatric disorders among CVD patients compared with their full siblings or matched population controls, by time of follow-up (<1 or ≥ 1 year from CVD diagnosis). CVD: cardiovascular disease. [a]Cox regression models, stratified by family identifier for sibling comparison or matching identifier (birth year and sex) for population comparison. Time since index date was used as underlying time scale. (e) Crude incidence rates (IRs) of different types of psychiatric disorder among CVD patients, their full siblings, and matched population controls, by time of follow-up (<1 or ≥ 1 year from CVD diagnosis). CVD: cardiovascular disease. (f) Crude incidence rates (IRs) for psychiatric disorders among different groups of CVD patients, their full siblings and matched population controls, by time of follow-up (<1 or ≥ 1 year from CVD diagnosis)[a]. CVD: cardiovascular disease. [a]We identified all cardiovascular diagnoses during follow-up and considered CVD comorbidity as a time-varying variable by grouping the person-time according to each diagnosis. (g) Crude incidence rates (IRs) and hazard ratios (HRs) with 95% confidence intervals (CIs) for psychiatric disorders among CVD patients compared with their full siblings or matched population controls, excluding CVD patients medicated with psychotropic drugs, by time of follow-up (<1 or ≥1 year from CVD diagnosis)[a]. CVD: cardiovascular disease. [a]CVD patients diagnosed during 2006–2016 were included in this analysis due to the availability of data on prescribed drug. In sibling comparison, 27.8% CVD patients and 23.5% siblings were excuded due to prior medicaiton of psychotropic drugs before index date. In population comparison, 30.6% CVD patients and 25.0% population controls were excluded due to prior medicated with psychotropic drugs before index date. [b]Cox regression models, stratified by family identifier for sibling comparison or matching identifier (birth year and sex) for population comparison. Time since index date was used as underlying time scale. Definition of psychiatric disorder included hospital visits as well as use of psychotropic drugs during follow-up. (h) Crude incidence rates (IRs) and hazard ratios (HRs) with 95% confidence intervals (CIs) for incident psychiatric disorder among CVD patients compared with their full siblings or matched population controls, restricting study period to 2001–2016 and excluding individuals with a history of alcoholic cirrhosis of liver or COPD, by time of follow-up (<1 or ≥1 year from CVD diagnosis). COPD, chronic obstructive pulmonary disease. [§]In sibling comparison, 1.14% exposed patients and 0.55% siblings were excluded due to a history of alcoholic cirrhosis or COPD before index date. In population comparison, 1.44% exposed patients and 1.01% population controls were excluded due to having a history of alcoholic cirrhosis or COPD before index date. [*]Cox regression models, stratified by family identifier for sibling comparison or matching identifier (birth year and sex) for population comparison. Time since index date was used as underlying time scale. Definition of psychiatric disorder included hospital visits as well as use of psychotropic drugs during follow-up.

- Supplementary file 2. Study design. CVD: cardiovascular disease. [*]67,745 families had more than one sibling affected by CVD.

- MDAR checklist

## Data availability

Data analyses were performed in STATA 17.0 (StataCorp LP). STATA script used in the primary analyses has been made available as supplementary appendix. Aggregated data used for generating figures are available in supplementary appendix. The original data used in this study are owned by the Swedish National Board of Health and Welfare and Statistics Sweden. The authors are not able to make the dataset publicly available according to the Public Access to Information and Secrecy Act in Sweden. Any researchers (including international researchers) interested in accessing the data can send request to the authorities for data application by: (1) apply for ethical approval from local ethical review board; (2) contact the Swedish National Board of Health and Welfare (https://bestalladata.socialstyrelsen.se/, email: registerservice@socialstyrelsen.se) and/or Statistics Sweden (https://www.scb.se/vara-tjanster/bestall-data-och-statistik/, email: scb@scb.se) with the ethical approval and submit a formal application for access to register data. The same contacts can be used for detailed information about how to apply for access to register data for research purposes.

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
