## [Editor Report]

Whether a diagnosis of cardiovascular disease (CVD) increases risks of psychiatric disorders is not well understood. Using a large population-based case-sibling study in Sweden, this important study suggests that patients diagnosed with CVD are at higher risk of psychiatric disorders, independent of familial factors shared between full siblings and comorbid conditions. While the analysis is solid, additional variables that relate to both CVD risk and mental health need to be incorporated in follow-up analyses.

---

## [Decision Letter]

**Decision letter after peer review:**

Thank you for submitting your article "Cardiovascular disease and subsequent risk of psychiatric disorders: a nationwide sibling-controlled study" for consideration by *eLife*. Your article has been reviewed by 3 peer reviewers, and the evaluation has been overseen by a Reviewing Editor (Prabhat Jha) and Eduardo Franco as the Senior Editor. The following individual involved in the review of your submission has agreed to reveal their identity: Rajini Nagrani (Reviewer #2).

The reviewers have discussed their reviews with one another, and the Reviewing Editor has drafted this to help you prepare a revised submission. Please review each comment carefully.

Essential revisions:

1) Address reverse causality, namely if a psychiatric illness (PI) developed before or about the same time as the CVD. The much steeper risk relationships early after a CVD event are so suggestive.

2) Could adjusting for matched factors in such cohort studies re-introduce bias into these estimates?

3) Address the point about the choice of preselecting exposure (CVD) group, rather than depicting the nationwide cohort of general population followed up for a disease outcome with each category having exposed and unexposed individuals.

4). Realign discussion such that the strength and limitations are the penultimate section of discussion along with reducing the definitive tone of the conclusions.

5). Authors are suggested to redo the flowchart such that the controls and cases are presented at the same level. At the moment it gives an impression that the comparison group (both population and sibling) is a subgroup of the CVD patients group.

6). Align the abstract, introduction, and discussion with the lack of lifestyle data, which would encompass several key potential confounders (notably smoking and alcohol use). Without it, it seems likely that there is a high risk of substantial confounding. It would be helpful if the authors adjusted the language to more accurately reflect this throughout, such as removing allusions to causal inference and tempering statements about adjustment for confounders where appropriate. This would include discussion specifically of alcohol and smoking risks in the Swedish population over time.

7). One of the main findings is that psychiatric disorder comorbidity diagnosed after cardiovascular disease diagnosis was associated with an increased risk of CVD-related mortality (such as shown in Figure 4). There needs to be more discussion of the potential relevance of cardiovascular disease or psychiatric disorder subtype in this relationship, both in the results and the Discussion sections. Intuitively, it would be expected that more severe cardiovascular diseases are associated with higher stress and a greater risk of psychiatric disorders. As such, the greater rate of mortality in those diagnosed with psychiatric disorders may largely be reflective of the initial cardiovascular diagnosis rather than having much to do with the psychiatric disorder diagnosis. Supplementary Figure 4 suggests that different subtypes of cardiovascular disease may have different relationships between psychiatric disorder diagnosis and survival rate, but this doesn't appear to be touched on at all in the results or the discussion.

8). Relatedly, could you please present similar analyses to Supplementary Figure 4 showing different survival curves by psychiatric subtype instead of cardiovascular subtype?

*Reviewer #2 (Recommendations for the authors):*

1. Authors are suggested to realign the discussion such that the strength and limitations are the penultimate sections of the discussion.

2. Authors are suggested to redo the flowchart such that the controls and cases are presented at the same level. At the moment it gives an impression that the comparison group (both population and sibling) is a subgroup of the CVD patients group.

*Reviewer #3 (Recommendations for the authors):*

1. As described in the public review, I feel the current version of the manuscript does not quite align at places in the abstract, introduction, and discussion with the lack of lifestyle data, which would encompass several key potential confounders. Without it, it seems likely that there is a high risk of substantial confounding. It would be helpful if the authors adjusted the language to more accurately reflect this throughout, such as removing allusions to causal inference and tempering statements about adjustment for confounders where appropriate.

2. One of the main findings is that psychiatric disorder comorbidity diagnosed after cardiovascular disease diagnosis was associated with an increased risk of CVD-related mortality (as shown in Figure 4). There needs to be more discussion of the potential relevance of cardiovascular disease or psychiatric disorder subtype in this relationship, both in the results and the Discussion sections. Intuitively, it would be expected that more severe cardiovascular diseases are associated with higher stress and a greater risk of psychiatric disorders. As such, the greater rate of mortality in those diagnosed with psychiatric disorders may largely be reflective of the initial cardiovascular diagnosis rather than having much to do with the psychiatric disorder diagnosis. Supplementary Figure 4 suggests that different subtypes of cardiovascular disease may have different relationships between psychiatric disorder diagnosis and survival rate, but this doesn't appear to be touched on at all in the results or the discussion.

3. Relatedly, could you please present similar analyses to Supplementary Figure 4 showing different survival curves by psychiatric subtype instead of cardiovascular subtype?

---

## [Author Response]

Essential revisions:1) Address reverse causality, namely if a psychiatric illness (PI) developed before or about the same time as the CVD. The much steeper risk relationships early after a CVD event are so suggestive.

Thank you for the comment. Previous studies have consistently reported an association between psychiatric disorders and CVD [1,2], thus, we agree that reverse causality may, in principle, explain some of the observed results indicating a rise in incident psychiatric disorders after incident CVD, particularly during the immediate period. Yet, it is reasonable to assume that a diagnosis of a lifethreatening disease, such as CVD, is in many cases a traumatic experience resulting in an immediate rise in risks of psychiatric disorders. Others have reported such associations e.g. after natural disasters and we have indeed observed such a pattern in our previous work, e.g., after cancer diagnosis [3]. However, we agree that reverse causality cannot be excluded and may partly contribute to the highly increased risk of psychiatric disorder immediately after CVD diagnosis. Indeed, some of these patients may have been attended for their psychiatric disorders in primary care before the incident CVD. As the Patient Register only captures in- and outpatient hospital care, we have conducted an additional analysis, also excluding individuals with previous prescriptions of psychotropic drugs (ATC codes: N05, N06) before their incident CVD – thereby adding a detection of patients with prevalent mental health problems attended by primary care. The results show similar point estimates (Supplementary File 1g) thus not supporting the notion that reverse causality is a major concern. Furthermore, the association is noted up to 28 years after CVD diagnosis, which is unlikely due to reverse causality.

We have now added our motivation for this additional analysis on the Method (Page 9), as below.

“Because the Swedish Patient Register includes only information related to specialist care, we might have misclassified patients with a history of milder psychiatric disorders diagnosed before index date attended only in primary care. To account for the reverse causality of having undetected psychiatric disorders or symptoms before the incident CVD, we performed a sensitivity analysis additionally excluding study participants with prescribed use of psychotropic drugs before the index date (ascertained from the Swedish Prescribed Drug Register including information on all prescribed medication use in Sweden since July 2005), and followed the remaining participants from 2006 to 2016.”

2) Could adjusting for matched factors in such cohort studies re-introduce bias into these estimates?

Thank you for the comment. Adjusting for matching factors should provide estimates with the same validity as using a stratified model. In our study, we matched individuals diagnosed with a CVD with their unaffected full siblings as well as 10 randomly selected, unexposed individuals, on the same age and sex, without such diagnosis. As controlling for matching variables is recommended when there are additional confounders [1,2], we used a stratified Cox model commonly applied in family-based studies [3,4].

References:

1. Sjölander A, Greenland S. Ignoring the matching variables in cohort studies – when is it valid and why? Stat Med. 2013 Nov 30;32(27):4696-708.

2. Mansournia MA, Hernán MA, Greenland S. Matched designs and causal diagrams. Int J Epidemiol. 2013 Jun;42(3):860-9.

3. D'Onofrio BM, Lahey BB, Turkheimer E, Lichtenstein P. Critical need for family-based, quasi-experimental designs in integrating genetic and social science research. Am J Public Health. 2013 Oct;103 Suppl 1(Suppl 1):S46-55.

4. Song, H., Fang, F., Arnberg, F. K., Mataix-Cols, D., de la Cruz, L. F., Almqvist, C., … and Valdimarsdóttir, U. A. (2019). Stress related disorders and risk of cardiovascular disease: population based, sibling controlled cohort study. bmj, 365.

3) Address the point about the choice of preselecting exposure (CVD) group, rather than depicting the nationwide cohort of general population followed up for a disease outcome with each category having exposed and unexposed individuals.

Thank you for the comment. As correctly pointed out by the reviewer, we used a matched cohort design, both in the population- and sibling comparison. We firstly identified a nationwide cohort of general population who were born after 1932 and were residing in Sweden 1987-2016. We then identified all exposed individuals with first-ever diagnosis of CVD and matched population controls from this same nationwide population.

A matched cohort design is applied here due to the strong confounding effects of some variables, e.g., age and sex, on the studied association between CVD and risk of psychiatric disorder. Exact matching on age and sex in our study makes the exposed and unexposed groups comparable and relief the confounding effects from matching factors in the design phase. Another practical viewpoint for why we use a matched cohort is a straightforward understanding of the comparison between exposed and unexposed groups being always at the same time, providing measures (such as risks and rates) during the follow-up period that are easily interpreted. Further, we have used this matched cohort design in many of our previous works [1,2] to maintain an identical design in both sibling and population comparison, so that the point estimates can be directly compared. The matched cohort design generates results of equal validity of the more conventional cohort design suggested by the reviewer [3] but has the additional quality of making the results from the various cohorts (here: population- and sibling comparison) more comparable. Our study therefore takes advantage of using a siblingcontrolled matched cohort, which is indeed a cohort design recommended for family-based studies [4] and provides results with similar validity as a full cohort.

We have now added a sentence and a reference in Method to motivate the use of matched cohort design (Page 7).

“We constructed a sibling-controlled matched cohort to control for the familial confounding according to guidelines for designing family-based studies.^24^”

We have now updated the flowchart to add a box in the top reflecting the source population where both groups were identified from, shown in Supplementary File 2.

References:

1. Song H, Fang F, Arnberg FK, Mataix-Cols D, Fernández de la Cruz L, Almqvist C, Fall K, Lichtenstein P, Thorgeirsson G, Valdimarsdóttir UA. Stress related disorders and risk of cardiovascular disease: population based, sibling controlled cohort study. BMJ. 2019 Apr 10;365:l1255.

2. Song H, Fang F, Tomasson G, Arnberg FK, Mataix-Cols D, Fernández de la Cruz L, Almqvist C, Fall K, Valdimarsdóttir UA. Association of Stress-Related Disorders With Subsequent Autoimmune Disease. JAMA. 2018 Jun 19;319(23):2388-2400.

3. Sjölander A, Greenland S. Ignoring the matching variables in cohort studies–when is it valid and why?. Statistics in medicine. 2013 Nov 30;32(27):4696-708.

4. D'Onofrio BM, Lahey BB, Turkheimer E, Lichtenstein P. Critical need for family-based, quasi-experimental designs in integrating genetic and social science research. Am J Public Health. 2013 Oct;103 Suppl 1(Suppl 1):S46-55.

4). Realign discussion such that the strength and limitations are the penultimate section of discussion along with reducing the definitive tone of the conclusions.

Thank you for the suggestion. We have now moved Strengths and Limitation section before the conclusion paragraph, as suggested (Pages 16-17).

We have now modified conclusions by removing definite terms such as “demonstrate”, “indicate”, and “increased”.

The modified conclusion:

Page 17: “Using a large population-based sibling-controlled cohort with up to thirty years of followup, we found patients with CVD are at elevated risk of newly diagnosed psychiatric disorder, independent of familial background shared between siblings, history of somatic diseases, and other cardiovascular comorbidities. The study further observes higher cardiovascular mortality among CVD patients with subsequent psychiatric comorbidities, providing evidence for increased surveillance of psychiatric comorbidity among newly diagnosed patients with CVD.”

5). Authors are suggested to redo the flowchart such that the controls and cases are presented at the same level. At the moment it gives an impression that the comparison group (both population and sibling) is a subgroup of the CVD patients group.

Thank you for the suggestion. Please note that the design is a sibling cohort but not a sibling case control study. We have now updated the flowchart presenting the matched population controls as well as unexposed siblings at the same level with their CVD patients, shown in Supplementary File 2.

6). Align the abstract, introduction, and discussion with the lack of lifestyle data, which would encompass several key potential confounders (notably smoking and alcohol use). Without it, it seems likely that there is a high risk of substantial confounding. It would be helpful if the authors adjusted the language to more accurately reflect this throughout, such as removing allusions to causal inference and tempering statements about adjustment for confounders where appropriate. This would include discussion specifically of alcohol and smoking risks in the Swedish population over time.

Thank you for the comment. We have now modified the text throughout, by removing causal terms.

In Abstract

Page 3, remove “halted causal inferences” and replaced with: “The association between cardiovascular disease (CVD) and selected psychiatric disorders has frequently been suggested while the potential role of familial factors and comorbidities in such association has rarely been investigated.”

Page 3, replaced “increased” with “higher”.

In Introduction, Page 5-6:

Removed “firmly”;

Replaced “a rapid rise” to “subsequent risk of”;

Replaced “attributed to” to “explained by”;

Remove “comprehensively”;

As the reviewer points out, we do recognize the potential unmeasured influence of lifestyle factors (e.g. smoking and alcohol consumption) on the studied associations as these data are not collected in registries. Sibling comparison can control for shared familial factors including lifestyles developed in early life. However, residual confounding from lifestyle factors not shared within siblings may still exist. In our study, the associations between CVD and psychiatric disorders were quite stable across calendar time, although somewhat stronger by the end of the observation period. The evidence does not suggest a drastic change in lifestyle factors in Sweden during the latter part of the observation period except for a slight increase in alcohol consumption [1,2] and liver cirrhosis [3]. Although we find it implausible that such underlying secular trends in lifestyle are a major contributor in the reported associations, we have now conducted additional analyses, excluding individuals with alcoholic cirrhosis of liver (ICD-10 code: K70.3) or COPD (chronic obstructive pulmonary disease, ICD-10 code: J44) as a proxy for heavy drinking or smoking. The results remained virtually unchanged.

We have now added reasons for stratified analysis by calendar years in Method (Pages 8-9):

“We performed subgroup analyses by sex, age at index date (<50, 50-60, or >60 years), age at follow-up (<60 or ≥60 years), history of somatic diseases (no or yes), and family history of psychiatric disorder (no or yes). We also performed subgroup analysis by calendar year at index date (1987-1996, 1997-2006, or 2007-2016) to check for potentially different associations over time (i.e., due to lifestyle factors that changed over time, including smoking and alcohol use).”

We found similar associations between first-onset CVD and incident psychiatric disorder with and without excluding individuals with a history of alcoholic cirrhosis of liver or COPD, used as a proxy for heavy drinking or smoking. The table has now added as Supplementary File 1h.

We have now added justifications in Method (Page 10) and in Discussion (Page 14 and Page 16-17), and as below: In method, Page 10:

“To account for potential impact of unmeasured confounding due to lifestyle factors, we performed a sensitivity analysis excluding individuals with a history of alcoholic cirrhosis of liver (ICD-10 code K703) or chronic obstructive pulmonary disease (COPD, ICD-10 code J44), as proxies for heavy drinking or smoking.”

In Discussion Page 14:

“The associations remained similar after excluding individuals with liver cirrhosis and COPD, as proxies for heavy alcohol consumption and smoking.”

Page 16-17:

“although we found similar results with and without excluding individuals with a history of liver cirrhosis or COPD, as proxies for heavy drinking or smoking (Supplementary File 1h). We did not have direct access to hazardous behaviors that could potentially modify this association, and therefore cannot exclude the possibility of residual confounding.”

References:

1. Statista. https://www.statista.com/statistics/693505/per-capita-consumption-of-alcohol-in-thenordic-countries/. Retrieved on 19 Aug.

2. Alcohol and Drug Report. Nordic Baltic Region. https://www.nordicalcohol.org/swedenconsumption-trends. Retrieved on 19 Aug.

3. Gunnarsdottir SA, Olsson R, Olafsson S, Cariglia N, Westin J, Thjódleifsson B, Björnsson E.Liver ;cirrhosis in Iceland and Sweden: incidence, aetiology and outcomes. Scandinavian journal of gastroenterology. 2009 Jan 1;44(8):984-93.

7). One of the main findings is that psychiatric disorder comorbidity diagnosed after cardiovascular disease diagnosis was associated with an increased risk of CVD-related mortality (such as shown in Figure 4). There needs to be more discussion of the potential relevance of cardiovascular disease or psychiatric disorder subtype in this relationship, both in the results and the Discussion sections. Intuitively, it would be expected that more severe cardiovascular diseases are associated with higher stress and a greater risk of psychiatric disorders. As such, the greater rate of mortality in those diagnosed with psychiatric disorders may largely be reflective of the initial cardiovascular diagnosis rather than having much to do with the psychiatric disorder diagnosis. Supplementary Figure 4 suggests that different subtypes of cardiovascular disease may have different relationships between psychiatric disorder diagnosis and survival rate, but this doesn't appear to be touched on at all in the results or the discussion.

Thank you for the comment. We agree that Supplementary Figure S4 indicates varying degree of compromised survival in relation to psychiatric disorders for different types of CVD.

We have now elaborated this in Results (Page 13) and Discussion (Page 15).

In Results, Page 13:

“The compromised CVD-specific survival differed by type of CVD, and was most pronounced for hypertensive disease, ischemic heart disease, and arrhythmia/conduction disorder compared to other CVDs (Figure 4 —figure supplement 2).”

In Discussion, Page 15:

“A compromised survival from CVD cause was indeed observed among most patients with common types of first CVD diagnosis and comorbid with psychiatric disorder, in particular among patients with hypertensive disease, ischemic heart disease, and arrhythmia/conduction disorder. Thus, particular clinical attention is needed for CVD patients with comorbid psychiatric disorders, particularly alcohol or drug misuse or psychotic disorders.”

8). Relatedly, could you please present similar analyses to Supplementary Figure 4 showing different survival curves by psychiatric subtype instead of cardiovascular subtype?

Thank you for the suggestion. We added, as suggested, a Figure 4 —figure supplement 3, showing the survival curves according to types of incident psychiatric disorder. A compromised survival was noted among CVD patients comorbid with non-affective and affective psychotic disorders, as well as alcohol or drug misuse.

We have now added this new analysis in Methods (10), Results (Page 13) and Discussion (Page 15).

In Methods, Page 10:

“We estimated the survival curves by types of first CVD types as well as by types of psychiatric comorbidities.”

In Results, Page 13:

“When studying categories of psychiatric disorders, we found that the compromised survival among CVD patients was confined to those with comorbid non-affective and affective psychotic disorders, as well as alcohol or drug misuse (Figure 4 —figure supplement 3).”

In Discussion, Page 15:

“Thus, particular clinical attention is needed for CVD patients with comorbid psychiatric disorders, particularly alcohol or drug misuse or psychotic disorders.”

Reviewer #2 (Recommendations for the authors):1. Authors are suggested to realign the discussion such that the strength and limitations are the penultimate sections of the discussion.

Thank you for the suggestion. We have now moved Strengths and Limitation section before the conclusion paragraph, as suggested (Pages 16-17).

2. Authors are suggested to redo the flowchart such that the controls and cases are presented at the same level. At the moment it gives an impression that the comparison group (both population and sibling) is a subgroup of the CVD patients group.

Thank you for the suggestion. Please note that the design is a sibling cohort but not a sibling case control study. We have now updated the flowchart presenting the matched population controls as well as unexposed siblings at the same level with their CVD patients, shown in Supplementary File 2.

Reviewer #3 (Recommendations for the authors):1. As described in the public review, I feel the current version of the manuscript does not quite align at places in the abstract, introduction, and discussion with the lack of lifestyle data, which would encompass several key potential confounders. Without it, it seems likely that there is a high risk of substantial confounding. It would be helpful if the authors adjusted the language to more accurately reflect this throughout, such as removing allusions to causal inference and tempering statements about adjustment for confounders where appropriate.

Thank you for the comment. We have now modified the text throughout, by removing causal terms. To explore the potential impact from lifestyle factors, we performed an additional analysis excluding individuals with a history of alcoholic cirrhosis of liver or COPD, as proxies for heavy drinking or smoking, and found similar results (Table added as Supplementary File 1h).

In Abstract

Page 3, remove “halted causal inferences” and replaced with: “The association between cardiovascular disease (CVD) and selected psychiatric disorders has frequently been suggested while the potential role of familial factors and comorbidities in such association has rarely been investigated.”

Page 3, replaced “increased” with “higher”.

In Introduction, Page 5-6:

Removed “firmly”;

Replaced “a rapid rise” to “subsequent risk of”;

Replaced “attributed to” to “explained by”;

Remove “comprehensively”;

In Method, Page 10:

“To account for potential impact of unmeasured confounding due to lifestyle factors, we performed a sensitivity analysis excluding individuals with a history of alcoholic cirrhosis of liver (ICD-10 code K703) or chronic obstructive pulmonary disease (COPD, ICD-10 code J44), as proxies for heavy drinking or smoking.”

In Discussion, Page 14:

“The associations remained similar after excluding individuals with liver cirrhosis and COPD (Supplementary Table S8), as proxies for heavy alcohol consumption and smoking.”

Page 16-17:

“although we found similar results with and without excluding individuals with a history of liver cirrhosis or COPD, as proxies for heavy drinking or smoking (Supplementary File 1h). We did not have direct access to hazardous behaviors that could potentially modify this association, and therefore cannot exclude the possibility of residual confounding.”

In conclusion,

Page 21: “Using a large population-based sibling-controlled cohort with up to thirty years of followup, we found patients with CVD are at elevated risk of newly diagnosed psychiatric disorder, independent of familial background shared between siblings, history of somatic diseases, and other cardiovascular comorbidities. The results further indicate higher cardiovascular mortality among CVD patients with subsequent psychiatric comorbidities, providing evidence for increased surveillance of psychiatric comorbidity among newly diagnosed patients with CVD.”

2. One of the main findings is that psychiatric disorder comorbidity diagnosed after cardiovascular disease diagnosis was associated with an increased risk of CVD-related mortality (as shown in Figure 4). There needs to be more discussion of the potential relevance of cardiovascular disease or psychiatric disorder subtype in this relationship, both in the results and the Discussion sections. Intuitively, it would be expected that more severe cardiovascular diseases are associated with higher stress and a greater risk of psychiatric disorders. As such, the greater rate of mortality in those diagnosed with psychiatric disorders may largely be reflective of the initial cardiovascular diagnosis rather than having much to do with the psychiatric disorder diagnosis. Supplementary Figure 4 suggests that different subtypes of cardiovascular disease may have different relationships between psychiatric disorder diagnosis and survival rate, but this doesn't appear to be touched on at all in the results or the discussion.

Thank you for the comment. We agree that Supplementary Figure S4 indicates varying degree of compromised survival in relation to psychiatric disorders for different types of CVD.

We have now elaborated this in Results (Page 13) and Discussion (Page 15).

In Results, Page 13:

“The compromised CVD-specific survival differed by type of CVD, and was most pronounced for hypertensive disease, ischemic heart disease, and arrhythmia/conduction disorder compared to other CVDs (Figure 4 —figure supplement 2).”

In Discussion, Page 15:

“A compromised survival from CVD cause was indeed observed among most patients with common types of first CVD diagnosis and comorbid with psychiatric disorder, in particular among patients with hypertensive disease, ischemic heart disease, and arrhythmia/conduction disorder. Thus, particular clinical attention is needed for CVD patients with comorbid psychiatric disorders, particularly alcohol or drug misuse or psychotic disorders.”

3. Relatedly, could you please present similar analyses to Supplementary Figure 4 showing different survival curves by psychiatric subtype instead of cardiovascular subtype?

Thank you for the suggestion. We added, as suggested, a Figure 4 —figure supplement 3, showing the survival curves according to types of comorbid psychiatric disorder. A compromised survival was noted among CVD patients comorbid with non-affective and affective psychotic disorders, as well as alcohol or drug misuse.

We have now added this new analysis in Methods (10), Results (Page 13) and Discussion (Page 15).

In Methods, Page 10:

“We estimated the survival curves by types of first CVD types as well as by types of psychiatric comorbidities.”

In Results, Page 13:

“When studying categories of psychiatric disorders, we found that the compromised survival among CVD patients was confined to those with comorbid non-affective and affective psychotic disorders, as well as alcohol or drug misuse (Figure 4 —figure supplement 3).”

In Discussion, Page 15:

“Thus, particular clinical attention is needed for CVD patients with comorbid psychiatric disorders, particularly alcohol or drug misuse or psychotic disorders.”